# Pathways to antibiotics in Bangladesh: A qualitative study investigating how and when households access medicine including antibiotics for humans or animals when they are ill

Patricia Jane Lucas[1]*, Mohammad Rofi Uddin[2], Nirnita Khisa[2¤], S. M. Salim Akter[2], Leanne Unicomb[2], Papreen Nahar[3], Mohammad Aminul Islam[2,4], Fosiul Alam Nizame[2], Emily K. Rousham[5]

1 School for Policy Studies, University of Bristol, Bristol, United Kingdom, 2 International Centre for Diarrhoeal Disease Research, (icddr,b), Dhaka, Bangladesh, 3 Brighton and Sussex Medical School, University of Sussex, Brighton, United Kingdom, 4 Paul G. Allen School for Global Animal Health, Washington State University, Pullman, Washington, United States of America, 5 Centre for Global Health and Human Development, School of Sport, Exercise and Health Sciences, Loughborough University, Loughborough, United Kingdom

¤ Current address: Faculty of Health, University of Technology Sydney, NSW, Australia
* Patricia.lucas@bristol.ac.uk

**Data Availability Statement:** Anonymised original data from the study can be accessed via the

## Abstract

### Background

To understand how to reduce antibiotic use, greater knowledge is needed about the complexities of access in countries with loose regulation or enforcement. This study aimed to explore how households in Bangladesh were accessing antimicrobials for themselves and their domestic animals.

### Methods

In-depth interviews were conducted with 48 households in one urban and one rural area. Households were purposively sampled from two lower income strata, prioritising those with under 5-year olds, older adults, household animals and minority groups. Households where someone was currently ill with a suspected infection (13 households) were invited for a follow-up interview. Framework analysis was used to explore access to healthcare and medicines.

### Findings

People accessed medicines for themselves through five pathways: drugs shops, private clinics, government/charitable hospitals, community/family planning clinics, and specialised/private hospitals. Drug shops provided direct access to medicines for common, less serious and acute illnesses. For persistent or serious illnesses, the healthcare pathway may include contacts with several of these settings, but often relied on medicines provided by

Loughborough University data repository at:
https://doi.org/10.17028/rd.lboro.8953118.v1.

**Funding:** This work was supported by the Antimicrobial Resistance Cross-Council Initiative supported by the seven UK research councils in partnership with the Department of Health, the UK Department of Environment Food and Rural Affairs and the Global Challenges Research Fund (Economic and Social Research Council grant ES/P004563/1 awarded to Dr Emily Rousham). The funder had no role in study design, data collection and analysis, decision to publish, or preparation of the manuscript.

**Competing interests:** The authors have declared that no competing interests exist.

drug shops. In the 13 households with an unwell family member, most received at least one course of antibiotics for this illness. Multiple and incomplete dosing were common even when prescribed by a qualified doctor. Antibiotics were identified by their high cost compared to other medicines. Cost was a reported barrier to purchasing full courses of antibiotics. Few households in the urban area kept household animals. In this rural area, government animal health workers provided most care for large household animals (cows), but drug shops were also important.

## Conclusions

In Bangladesh, unregulated drug shops provide an essential route to medicines including those prescribed in the formal sector. Wherever licensed suppliers are scarce and expensive, regulations which prohibit this supply risk removing access entirely for many people.

## Introduction

Antimicrobial resistance (AMR) threatens our ability to treat infectious diseases, and has been identified as a global threat to health [1]. Low and middle income countries (LMICs) already experience the worst impact, with highly resistant organisms already seen in the environment [2–4]. The overuse and misuse of antibiotics is a recognised driver of resistance [4, 5]. In order to reduce this hazard, the rational and responsible use of antimicrobials (AM) is identified as a key element of global actions to protect the efficacy of AM and contain AMR [6–8]. Stewardship policies aim to achieve this by improved prescribing practices, using antibiotics only when needed and with correct dosage and duration [9]. These policies need to respond to the particular healthcare context and recognise professional and cultural norms [10].

However, in LMICs the paradox of both limited access and excess use of antibiotics is recognised [4]. In Bangladesh, for example, falls in childhood mortality from pneumonia and post-natal sepsis have been partly attributed to unregulated, easy access to antibiotics [11]. Common infectious diseases still contribute to high childhood mortality [12]. At the same time, overuse of antibiotics has been widely reported, often attributed to purchase in 'drug shops' [2, 11, 13–15]. Although there are regulated pharmacies in Bangladesh, most drug shops are unregulated vendors selling medicines and are present in even very remote communities [15, 16]. Eighty to ninety percent of drug shops operate without a drug license [16]. Drug shops fill an important gap in healthcare infrastructure in Bangladesh and are reported to be the preferred first point of contact for most of the population because of the low cost, long opening hours and wide geographic spread [15, 17]. In a recent study, Ahmed et al [18] found that among children presenting at a tertiary hospital with diarrhoea, of those who had already taken antibiotics at home only 6% had been prescribed by a qualified provider. One study observing practice in 75 drug shops found that antibiotics were the highest selling medicine [13].

Furthermore, in countries such as Bangladesh with a highly plural health system [19] and limited access to formal healthcare, the challenge is to understand how it might be possible to reduce inappropriate use of antibiotics, including sub-standard drugs and inadequate doses [11, 20]. To achieve this we need to understand the needs and behaviours of all the actors in the system, including an understanding of how both preferences and availability of healthcare determine access to antibiotics. Up to now, most research into people's healthcare seeking and

medicine access in Bangladesh has been undertaken in drug shops or healthcare settings. Moreover, food-producing and domestic animals have been relatively neglected in relation to human health and AMR [21]. The closely shared environments of humans and animals has been shown to have high potential for direct or indirect transmission of resistant bacteria [22]. We need to establish the different pathways that families follow to reach antibiotics for themselves and their household animals [2].

This study is one part of a larger study using integrated research to inform the development of interventions to reduce the risk of antimicrobial resistance in Bangladesh [23]. The element reported here aimed to explore how households in Bangladesh access antibiotics for themselves and their domestic animals.

## Materials and methods

Semi-structured interviews were selected to capture experiences, beliefs and preferences about medicine access and use. The protocol for the full study including selection of study sites, selection and recruitment of households, our approach to translation of materials and analysis is published in full elsewhere [23]. In brief, we selected one urban area (within Gazipur District) and one rural area (within Mirzapur District) not previously involved in community research to our knowledge. We purposively sampled households meeting inclusion criteria of i) above and below a household income cut-off (see Table 1, ii) including children under 5 and/or older adults residing reflecting the greater susceptibility to infectious disease and high use of antibiotics among young children [24] iii) Food-producing animals or livestock in the home compound. In Bangladesh, keeping animals in the home compound for food or commerce and therefore sharing their living environment with families is common [22, 25]. Household income cut-offs were based on rural and urban mid-range estimates of household income adjusted for inflation [26], focusing our research on two strata at the lower end of the income distribution. We also aimed to include minority indigenous population in rural areas to maximise variation in the social context of the sample [23]. We aimed to interview both those who take care of people or animals when they are sick (care takers) and those who makes decisions for the household (decision makers). Where someone in the household currently had symptoms suggestive of an infection (e.g. fever or diarrhoea), we requested to return in two weeks for a follow up interview to get a contemporaneous account of healthcare seeking and medicine use. We continued recruitment until our data saturation was reached, which we judged against our sample targets and in discussion with the field researchers to establish when they felt they were no longer hearing new material.

**Table 1. Household interviews: Characteristics of sample.**

|  | Rural (n = 24) | | Urban (n = 24) | |
|---|---|---|---|---|
|  | ≤15,000 Taka* (n = 12) | >15,000 Taka (n = 12) | ≤20,000 Taka (n = 12) | >20,000 Taka (n = 12) |
| Caregivers | 6 females | 6 females | 6 females | 6 females |
| Decision makers | 2 females, 4 males | 3 females, 3 males | 3 females, 3 males | 2 females, 4 males |
| Livestock in the home | 12 | 12 | 0[+] | 2[+] |
| Households with child <5 years | 9 | 10 | 9 | 9 |
| Households with older family member† | 6 | 6 | 5[+] | 6 |
| Households with a currently ill member | 3 | 3 | 3 | 4 |
| Indigenous ethnic households | 2 | 2 | 0 | 0 |

*Monthly household income in Bangladesh Taka. 15,000 Taka~US$178, 20,000 Taka~US$237

[+]Under recruitment relative to target

†Women aged >62 years and men aged >65 years old; eligibility threshold to receive government old age allowance in Bangladesh.

Approval for the study was provided by Institutional Review Board at the International Centre for Diarrhoeal Disease Research, Bangladesh following review by the Research and Ethics Committees (PR-16100) and from Loughborough University (R17-P081). Households were recruited and interviewed by local researchers who visited house to house using a screening questionnaire which include questions about household composition, income and ethnic group. Our team included men and women and a researcher from a minority ethnic group whose first language was not Bangla. Households were provided with written study information in Bangla which was read aloud to them and they were given an opportunity to ask questions. If they agreed to be interviewed, they were asked to sign a written informed consent form in Bangla. Data were held on a secure, password protected server and anonymised prior to analysis.

Our interview guide was developed in collaboration with the wider research team and piloted before use, and is available in English and Bangla versions at https://doi.org/10.17028/rd.lboro.9333158.v1. The guide was reviewed as interviews progressed to ensure questions were appropriate and to explore emerging themes of interest. We explored the current health of family members and domestic animals, recognition of illness, health care decision making, preferences for healthcare providers and medicines, antibiotics and their side effects including antibiotic resistance (AR). The terms antibiotics and AR were explained where needed by field staff. We began by using these terms and asking if they were known, but where people did not know the terms they explained further using common language (see interview guide), provided examples of antibiotic names and packaging, provided leaflets published by icddr,b and asked to see any medicines in the home to identify any antibiotics currently being used. We asked about vulnerable household members and tried to disentangle who made decisions, who consulted health care providers, and who cared for both people and animals when they were ill. In follow up interviews we asked about progress of illness and medicines purchased and consumed since our previous visit.

Interviews were audio recorded and transcribed in full in Bangla. We considered our approach to translation carefully [23] and used an approach which kept first stage analysis in Bangla in most cases. A third of interviews were translated in full and these were used to develop our analysis plan. The field team were asked to identify rich interviews from both rural and urban areas for this purpose. For those interviews not translated in full, the bilingual research team members coded Bangla transcripts into framework charts, and these charts were then translated into English for sharing with the full team for further analysis. As analysis progressed, we checked the translated material against the Bangla transcripts where needed.

Framework analysis was used [17, 27] which asks researchers to code data into a framework, a process known as charting. The team of qualitative researchers (MRU, SMSA and NK, our field team based in Bangladesh with a mix of public health and sociology training, led by PJL, a psychologist based in the UK and FAN, a sociologist based in Bangladesh) together developed a draft framework using the translated interviews. All members of the qualitative research team except PJL were bilingual in Bangla and English. The draft framework was shared with the wider study team (which included social anthropologists, microbiologist and public health specialists) and all worked together to iterate versions until a final version which allowed capture of all information was agreed. Four transcribed and translated interviews were independently coded by MRU, SMSA, NK and PJL and compared to ensure consistency. MRU, SMSA and NK undertook the remaining coding with guidance and advice from PJL and FAN, and translating Bangla data into English as needed. Fifty percent of interviews were double coded for quality assurance purposes. First stage analysis of frameworks was undertaken in a team setting with PJL, FAN, MRU SMSA and NK. Further analysis was completed by PJL and MRU. All materials were shared with all team members for comment. Pathways to

medicines were summarised and then categorised by the team, paying particular attention to access to antibiotics. We looked at knowledge and understandings of antibiotics, and themes emerging in these.

## Results

We approached a total of 49 rural and 67 urban households. In the urban sites, we struggled to find households with animals and therefore we relaxed this criterion. Of the 67 households approached, 36 (54%) households did not meet inclusion criteria. Of the 31 eligible households, seven (22.5%) refused to participate and we interviewed the remaining 24 households (77.5% of those eligible, 36% of those approached). In the rural site, of the 48 households approached 22 (46%) did not meet inclusion criteria, and in two households neither the decision maker nor carer were at home. We interviewed the remaining 24 households (100% of those eligible, and 50% of those approached) Table 1 reports the household sample, full details are reported in S1 Table. Interviews took place in the period May 2017 to January 2018. In total, we undertook 61 interviews (48 households, 13 follow up interviews). Themes and findings are illustrated using quotes from individuals, identified using the notation R/U (rural or urban), F/M (male or female) and a household number, i.e. RM304 refers to a man from household 304 in the rural area.

We use the term medicines throughout as the term was used in the interviews, describing any substances consumed or injected to improve the health of people or animals. This includes vitamins, tonics, homeopathic remedies and other substances that would not be viewed as active treatment in allopathic medicine. We report findings specific to antibiotics where we were able to discern them.

### Pathways to medicines for humans

The individual stories of where and when sick individuals sought and received medicines often encompassed a series of consultations in a number of settings. However, five types of settings through which allopathic medicines or treatment recommendations could be sought were identified in our interviews: Drug shops, private clinics, government and charitable hospitals, community /family planning clinics, private and specialised hospitals. In addition, seven households reported visiting traditional healers or *fakirs* but as these do not provide allopathic medicine, these are not described further here. Each setting is described below together with examples of how they were used to access medicines.

### Drug shops

Drug shops are the common name for retail locations which sell allopathic medicines, regardless whether they are licensed [16]. Visits to the drug shops were common to all participants, and most reported receiving treatment advice as well as purchasing medicines. People often referred to the drug sellers and drug shop owners as 'doctors' although they are unqualified and may have simply inherited the business. Some refer to them as "small doctors" or "village doctors", differentiating them from "big doctors" or "MBBS [Bachelor of Medicine, Bachelor of Surgery] doctors".

Drug shops were visited for less serious illnesses where an immediate supply of medicine was expected. This was the most common direct route to medicines:

> "Actually for small types of disease we go to this market where they have small doctors, we usually go to them for our primary health care." RM304

"Most of time we visit the [drug shop] seeking primary treatment like for gastric, fever and cold and we take medicines from there." UM201

Drug shops give advice and medication even where the sick person is not present, when symptoms would be described by someone else who was able to visit the shop:

"I send my mother-in-law and she buys medicine so I don't need to go. . .I mean he [name] is an experienced doctor and also sells medicines so he gives what medicines we need regarding our symptoms." RF301

Drug shops also provide an indirect route to medicines, where people collect prescriptions or recommendations from other healthcare providers then buy these from the drug shops:

"Most of time I visit to [drug shop] near our house for purchasing medicines and sometimes collect the medicine from [larger shopping area]. . .When I was suffering with high pressure problem, then the doctor prescribed me medicines for regular intakes and I purchase it from [drug shop] near our house." UF207

## Private clinics with qualified doctors

Private clinics range in size from an individual with an MBBS only operating from a room (a "doctor's chamber"), often attached to a drug shop to quite large clinics with diagnostic facilities and a few inpatient beds. The term doctor's chamber refers simply to a private consultation room and often these are village/unqualified doctors, but in this section we only include those examples where participants confirmed to us that a MBBS/qualified doctor did practice. Qualified doctors may move between locations or work elsewhere in the daytime and only attend the clinic/chamber for a few hours, but these small clinics and chambers can often be found close to home. Here one urban participant tells us about a smaller and larger private clinic they visit:

"Nearby like [shopping area] there is a MBBS Doctor [name], he may sit [works] at another place but he sits [works] here in the evening. We have known him for many years so we go to him most of the time." And "He is a paediatrician and those who are pregnant they go to a nearby doctor of [name of clinic]." UF306

It is expensive to see a MBBS doctor in a private clinic:

"Close to us there is a chamber inside the drug shop and he sits [works] there. He sits at 6 pm and prescribes from there. If I spend seven hundred Taka [around US$8] to get the doctor's advice then how I will purchase medicine for the patient? For ten days medicines we have to pay one thousand Taka [US$12], I brought it for my daughter. If I have to pay seven hundred Taka to the doctor and one thousand for medicine then how can I pay for the medicine? We are poor people." U104

## Government/Charitable hospitals

In the urban area there was a large government hospital and in the rural area a large charitable hospital nearby. These are presented as a common route since people would not have access to both and they provide similar services. In both locations people were able to access generalist and specialist doctors, diagnostic facilities and inpatient beds but a very long wait to see a

doctor was expected. People described visiting these hospitals when someone was seriously ill, and where treatment taken directly from drug shops has failed. For example, a woman in the rural area (RF303) told us "when we become sick with serious illness then we go to [charitable hospital]". She named three occasions she had visited this hospital: when her father-in-law was sick and needed "six types of medicines"; when she had severe pain she went for tests and was diagnosed with a urine infection, and when her new-born child was diagnosed with pneumonia and admitted for a 10 day stay.

The government hospital charges minimal fees to visit (10–50 Taka, <50 US cents) and could provide free medicines from a restricted list. Where patients needed other medicines or when the free supply had run out (which happened often) they were given a prescription to take to a drug shop. The charitable hospital charged a very low fee for consultation and provided no medicines but instead a subsidised price was offered in certain pharmacies with a hospital prescription. In this way, both of these settings relied on drug shops for medicine supplies.

### Government community/Family planning clinics

In addition to government hospitals, some rural households reported visiting community and family planning clinics (in the Upazila health complex). These provided some healthcare although they were not staffed by qualified doctors and services were very restricted. Only medicines from a restricted approved list were available, and these had often run out and were not available. Patients therefore took prescriptions to a drug shop:

> "In maximum time [most of the time] I visit the Community Clinic. They provide us medicine in exchange for two Taka (2 US cents). . .They provide vitamin, medicine for fever, gastric, diarrhea. " RF107

However, they were not a reliable source of medicines and patients must buy the medicines needed from a drug shop. As another participant told us:

> "We go less often to that community clinic because most of the time there is a shortage of medicine. At the beginning of the month you can get medicine but by the end of the month they can't give you medicine. . .Demand is very high so that they can't provide for the whole month." RM304

### Specialised hospitals

A very large range of general and specialised hospitals were named which our participants used rarely for particular health problems. These were a mix of private, charitable and government provided. Participants in our sample had used a Tuberculosis hospital, an eye hospital, a diabetes hospital, or in two cases visited the icddr,b (*International Centre for Diarrhoeal Disease Research, Bangladesh)* hospital some distance away in Dhaka city. For some this was the last stage after unsuccessful attempts at treatment elsewhere, but also in emergencies. One visit to the icddr,b hospital was for an emergency visit for a baby with diarrhoea and a very high temperature. One patient had visited one of the largest and most expensive hospitals in Dhaka. Private hospitals provided medicines to those attending.

### Multiple routes through pathways

While these five routes describe the types of settings used, only the drug shops operated as a regularly used standalone pathway. Serious, chronic and persistent illness resulted in multiple

contacts, escalating costs and increasing use of specialisms. A typical account of a serious illness was given by a Hindu man in the rural area:

*"No, for her eye problem I went to the [charitable] hospital (two times) then I went to the [neighbouring bazaar] when they couldn't do anything so I went to the [private clinic] but they didn't give any medicines and only said to go to the Dhaka Eye Hospital."*RF307

In this way, drug shops in bazaars were operating as the first, intermediate and final points of access to medicines for most households.

## Differences by household income, location and decision-maker

We were interested to know whether pathways varied by household or participant characteristics, to the extent possible in this qualitative study. With the exception of community/family planning clinics which were only present in the rural area, all pathways were used in both rural and urban areas, and across income groups. Outside of differences in distances travelled, rural and urban households described largely similar approaches to accessing medicines. Unsurprisingly, more of the poorest household mentioned using government or charitable facilities but higher income households were also concerned with price and this determined choices made:

"We go there because the price is reasonable on the other the hand private clinic is very expensive; that's why we go to [Charitable] hospital." RM304, higher income.

The main difference between male and female participants, was in who made decisions about health care treatment seeking and purchase of medicines. All but one of the men we spoke with either made decisions independently or in collaboration with other adults usually their brother or mother but also neighbours. One man (UM101) told us he consulted his wife because she takes care of the baby and knows her condition. In contrast, women were frequently having to defer to the decisions of others even when this meant waiting for some days for a decision (RF108). These could be husbands, brother-in-law, mother-in-law or older sons:

"Most of the time I take the treatment seeking decisions because I am the guardian of this family". UM201, male

"Decisions are mainly made mutually, not only by me but also mutually." RM304, male

"My father-in-law and husband jointly take treatment decisions, they consult each other. But, most of the time my children's father decides". RF207, female

"I asked from [child's father]. I asked like this, he [son] is feeling sick now what can we do? Then he said go to the doctor. . .This decision was made by [child's] father" [her husband decides which doctor and pharmacy to visit, and she will usually go too and they will decide what to purchase. If her husband doesn't accompany her] "For decision sometimes I make a call to [child's] father and tell him the doctor wants to give this medicine." UF306

Nonetheless some women were making decisions independently, including when their husbands were working abroad, or making decision alongside their husbands:

"I take the treatment and make decisions by myself and also share it with my husband." RF203

The four households (Rural F204, M205, M304 and F305) from minority indigenous groups included two lower and two higher income, two male and two female participants. One of

these households (RM205) had relatives (from the same community) who were drug sellers and they visited their shops by first preference for both advice and purchase of medicines. Otherwise their stories were very similar to others we interviewed in the rural area:

"How can I say, everyone goes to him [village doctor]. . .We always go to him and we get well quickly after taking his medicine." RF205

### Pathways to medicines for household animals

We found only two urban household with any animals (UF309 kept pigeons and UF202 chickens, both higher income). In the rural area most households (16 in total) had both cattle and chickens in their home compound. Cattle were the most commonly kept animals (22 households), 18 households kept poultry (chickens, ducks, pigeons) and 1 household also kept a goat. There were fewer pathways to medicines for domestic animals: drugs shops, government services and private practitioners. Although there were fewer pathways, in this area, the pathways were more robust because of the existence of a qualified veterinarian in the local government office who was highly motivated to improve local practice to benefit poor families and their domestic animals.

### Drug shops

Householders visited the local bazaar and bought medicines directly from the drug shops. Some drug shops sold only animal medicines but others sold both human and animals medicines. Similarly to humans, this was a straightforward and direct route to medication:

"We purchase animal medicines from the [local] Bazar." RF203

### Government provision

In rural areas in the local Union offices (the Parishad) there was a livestock office which included, Artificial Insemination (AI) officers, and a qualified veterinarian overseeing all activities. The AI officers would have received training in AI initially under a government scheme to support small farmers, but they seem to be used for a greater variety of tasks and may have received more specific training for these. Through this office and its staff, households did not pay for consultations, could access some free medicines in the offices, and could directly access medicines at a competitive cost. The provision was used particularly for cattle, which are a very valuable commodity:

"If (we) need any medicine we go to government veterinary doctor [AI technician]. Sometimes we call them to our house; they give prescription or medicine whatever the disease demands. After that gradually the animals are cured. We always go to government doctors." RM304

### Private practitioners

A few households used private veterinary practitioners who they either contacted by phone or where they had chambers in a local drug shop, paying a consultation fee in addition to the cost of medicines. It was not clear in our interviews whether these were qualified practitioners even when we asked, there was not the same recognition of qualified Veterinarians as of MBBS doctors:

"We have the animal doctor's phone number when we need him we just make a phone call, then he comes with medicines and gives treatment and gives medicines to the animal. . .He has a pharmacy shop and chamber where he sits, when we call him he bring medicines from his shop and gives it to the animal." RM103

These practitioners provided medicines or "pushed" (injected) medicine themselves. One participant explains that she uses an "animal doctor" for her cows but the drug shops for chickens:

"When we call the animal doctor, they bring medicines with them. . . he plays both roles doctor and drug seller so that we don't have to go far from this area. . .For chickens' medicines, we get it from the market." RF301

One participant told us he used to visit an unqualified animal doctor, but now *"if any animal gets severe diseases then I call the big doctor (Veterinary doctor from Upazila livestock office)."* RM104

We also asked households about use of human medicines for their animals. Only one participant reported this practice. One rural woman (RF305) reported giving "napa" (paracetamol/acetaminophen) which she also called "fever medicine" mixed into rice or water to her chickens when they were sick.

## Pathways to antibiotics

People's knowledge and understanding of antibiotics could be classified under 4 themes: knowledge by name, little or no knowledge, powerful medicines, and knowledge of action.

The name "antibiotic" was recognised by many in connection with human medicines, although it is unclear if the majority of people understood what this meant. Similarly, the brand names of particular antibiotics were often provided when people described particular illness episodes and the treatment received but this did not necessarily imply knowledge that they were antibiotics.

Antimicrobials for humans named by participants or shown to us were Amoxycillin (known as Fimoxyl syrup), Ciprofloxacin (Ciprocin), Azithromycin (Thiza), Metronidazole (Amodis), Cefixime (Roxim, Orcef), Cefaclor (Ceflon, Navacef), and Fluconazole (Flugal). In the smaller number of households with animals, participants were less clear about when antibiotics had been given to animals, and none given to animals were mentioned by name, probably because these were more often administered by a practitioner.

Understanding of antibiotics was often limited to the word, or the dosage pattern. One participant knew that that Fimoxyl and Ciprocin were antibiotics, but:

"I do not understand so much about antibiotics. I am not as educated to understand antibiotics. Everybody says that antibiotics are good and cure diseases and I also know about that. . .Everybody called it antibiotic, we also called it as antibiotic. The Doctor also called it antibiotic. It has never come to mind why it is called antibiotics. . .We didn't ask them so they did not say anything." RM101

Often people could not offer any ideas of what an antibiotic was, or had incorrect knowledge:

"all medicines are antibiotic medicines. . . .it seems that all are like this, such as- Napa (paracetamol/acetaminophen) is an antibiotic and for pain [the doctor] gives antibiotic medicines." UF01

When probed, antibiotics were recognised as big or powerful medicines, a medicine that is stronger than other medicines, or that is used when other medicines have failed:

"Now it is necessary to take powerful medicine or when the fever isn't cured by the normal medicine like Napa." RF08

"You know when we can't get well from fever and maybe the fever is in another stage then maybe the doctor gives us antibiotic." RF302

One participant told us that antibiotics acted more slowly in the body, but more often they were recognised as faster acting than other medicines. We also noted that several people believed that because this was strong medicine, it could cause harm to the body and should be used with care:

"Normal drugs will take four to five days to cure the disease. But an antibiotic takes only two days. . .But at the same time it's also worse because after taking this medicine, some problems may happen.

Interviewer: What kinds of problem may happen?

Participant: The body will become weak because it has a lot of power. . .The body will become weak, the head dizzy, and someone can face vomiting problems. So many problems may happen." UF206

Few participants identified or spoke about antibiotics as a discrete class of medicines and some had misunderstandings. Although a few people mentioned germs as a cause of illness, one person (UF202) thought antibiotics treated only viral diseases. Just one woman correctly understood how an antibiotic works:

"Interviewer: I want to know your perception . . .you know Thiza because you have given this to your child?

Respondent: Maybe it kills harmful bacteria quickly that's why the doctor prescribed this medicine." RF301

The lack of recognition of a specific group of medicines or knowledge that antibiotics are used to treat bacterial infections means we could not reliably distinguish whether antibiotics were accessed through each pathway. Nonetheless, easy access to antibiotics via drug shops for humans and via local veterinary staff for animals was reported.

The main barrier to use was not access, but price. Antibiotics were reported to be significantly more expensive than other medicines, and indeed the high cost was a recognisable feature of antibiotics:

"The price of antibiotic is just double of normal medicine." RF08

"this medicine, it is high priced one tablet is 35 Taka or 40 Taka (US$0.5). That means it is valuable medicine and I understand that it is antibiotic medicine." UF303

When one drug shops did not have a particular medicine, people reported visiting another drug shop (even in the rural area there are several drug shops in the bazaar) or that the drug shop seller would get the medicines they wanted for them:

**Table 2. Antibiotics use for current illnesses.**

| Household | Who was ill | Treatment at first interview | Follow up interview |
|---|---|---|---|
| RF106 | One-year old grand daughter with a fever. | She was improving, her father had given her a syrup from the drug shop for 2/3 days. | Confirmed the child had received antibiotics to take three times a day for three days. When the child improved, they threw away the remaining medicines because they were concerned if they kept them the child might eat them and it would do her harm. |
| RF301 | 3-year old son with a fever and vomiting. | Her mother-in-law visited the drug shop, and bought Ace, Histacin and Thiza (Azithromycin group, 30 ml syrup). | She had taken the antibiotic, which was prescribed once a day for three days. But the child improved after only one or two days so they didn't give the remaining medicine. |
| UM201 | Four-year old twin daughters fever with cough. | One twin became ill a couple of days earlier, then the other. The father had bought a syrup (an antibiotic) from the drug shop, but they are not improving so he was considering visiting the Government doctor. | They had visited the Government doctor, who confirmed the child also had scabies for which an ointment was prescribed. The father had shared the antibiotic observed at the first appointment between the twins. He said that their symptoms were improved, and he couldn't afford to buy additional medicines so that both daughters could have a full course. |
| UF202 | Three-year old son with diarrhoea. | Admitted to Government hospital the day before and put on a saline drip. | The Government doctor prescribed a 5 day course of antibiotics. She purchased only 3 days of the antibiotic for her son from a drug shop. |
| UF01 | Four-year old grandson with fever. | Was very ill the night before, and had collapsed. She had intended to go to the Government hospital, but met a drug seller on the way who told her his medicine would cure her grandson. She bought this to avoid a 2–3 hour wait at the hospital. | She confirmed that she had given her grandson a medicine twice a day for 5 days and another 3 times a day for three days. Researcher confirmed these were antibiotics (participant was unsure). She had thrown away the bottles when completed. |
| UF302 | Ten-month old son with a cough and vomiting. | Had been ill for 8 days and already received antibiotics. The doctor (qualified) said his cold was "very severe" and gave 3 syrups for 7 days (including an antibiotic) plus a nebulizer and said to return when these were finished. She was advised to give "pure water" and Ace syrup for the fever. | The child was still unwell (and the interview was cut short for this reason). Since our last visit the same doctor said the cold had "become an infection" and provided 2 more medicines, at least one of which was an antibiotic. He also said that if the child didn't improve he would have to inject medicine. |
| UF307 | Seven-month old daughter with a fever and cough. | She had received 7 days medication from the drug shop. After 2 days with no improvement she returned to the drug shop and received 3 syrups (unclear whether these included an antibiotic). After 3 more days her daughter improved and she returned to the drug shop who gave 4 types of medicine including an antibiotic (7 days). The fever was improved. | Since the last interview the child also became constipated. Her husband went to the doctor (unqualified) and he sold two syrups and a suppository. She didn't know what medicines she had been given, but this included 4 syrups including one where she was told to give two spoonfuls in the morning and two in the evening. |
| UF309 | Two children (her son and her niece) had a cold with fever, and also an allergic rash on their skin (a "stain"). | They had visited the doctor and purchased antibiotics to begin that day. | She gave the antibiotics to both of the children every day for 7 days and Flugal for the rash, and they got better. Her son has another cold, but it is "not a big illness" and he doesn't need any more medicines. |

"If there is no medicine available there, then the pharmacy will collect it and provide among them after one or two days later." RF06

The ease of access to antibiotics was illustrated in the 13 households where we followed up suspected active infection at first interview. These were mostly children with fever. In total eight of these households reported receiving antibiotics over the period of the illness, and Table 2 provides a summary of these cases. In these households, accessing medication had not been difficult and decisions about treatment were determined by the resolution of symptoms alone. Two of the stories provided exemplify this. In one wealthier urban household (UF302), a 10 month old baby with a cough had been taken to a qualified doctor before our first visit. The doctor said that this cold "is in the very severe stages" and gave 3 syrups (including an antibiotic) and a nebulizer. At the follow up visit the child still looked sick, and the doctor had

given two more antibiotics (one for 10 days and another for 7 days) and had told the family that if the baby didn't respond he would "push" (inject) antibiotics.

In a poorer rural household (RF106) with a one-year-old with a fever, the family had visited a drug shop seller (who they called doctor) who gave her antibiotics to take three times a day for three days. In the follow up interview, they told us they had stopped the medicine when the baby improved and threw away the remaining drugs because they were concerned if they kept them in the home the children might eat them.

## Discussion

This study examined the routes by which lower income households in Bangladesh accessed antibiotics when they or their animals were ill. In both urban and rural areas people accessed allopathic healthcare through five settings: drugs shops, private clinics, government/charitable hospitals, community/family planning clinics (rural area only), specialised and private hospitals. For less serious and acute illnesses, drug shops were often used. For persistent, chronic or serious illnesses multiple settings were used in combination depending on their availability and the length of illness. However, most pathways route through drug shops, which both provide immediate treatment and provide access to the medicines not available or provided by others. Pathways to antibiotics were common to all participant groups, and we found no clear differences by area (urban/rural) or household income or in the minority community. Both the word antibiotic and the brand names of individual antibiotics used by people were often known, but most people understand them as powerful, valuable (and expensive) and highly effective medicines without understanding them as a class of medicines used to treat bacterial infections.

Few urban households kept animals, probably because many lived in apartments with no outside area. In the rural area, government trained and paid animal health workers provided care for animals supplemented by medicine purchase at drug shops.

This study agrees with research on drug shops. In a survey of pharmacies (including regulated shops with qualified pharmacists), Saha et al [13] report that antibiotics were regularly provided without proper history taking, without proper prescriptions and in inadequate doses. Chowdhury et al [24] reports that 38% of those attending sought medicines for someone other than themselves and that 90% of visits were first point of care visits. Like us, the authors report in their 5-day follow up telephone interviews that medication ceased when symptoms improved and that further treatment seeking was common when symptoms were not improving quickly.

For the global challenge of antibiotic stewardship, this work illustrates the difficulty of reducing excess use of antibiotics without restricting access. In our study everyone we spoke to relied on drug shops to access the medicines that they need including where prescriptions were provided. Unregulated drug shops are plentiful and provide antibiotics to large numbers of people [2]. But their workers are mostly untrained, they provide antibiotics that may not be needed, and without proper attention or knowledge of dosage [2]. A country-wide survey of drug shops in Bangladesh reported that only 35% of drug shop customers have a prescription, and only rarely did drug shops require a prescription to purchase antibiotics [16]. Reducing this kind of unregulated access to antibiotics is a clear target for stewardship campaigns at a global level. Indeed, in April 2019 the High Court in Bangladesh directed Government to outlaw the sale of antibiotics without a prescription, in response to concerns about antimicrobial resistant infections [28]. However, actions of this type assume that prescriptions from qualified physicians are a) available and b) compliant with antimicrobial stewardship guidance, but both of these assumptions can be challenged.

Given the level of morbidity and the size of the qualified workforce, it is not plausible that people in Bangladesh will be able to attend qualified physicians for all their illnesses [29]. The shortage of qualified doctors particularly outside of urban areas is well recognized [29]. In our study although qualified doctors were consulted by many, they were seldom consulted first because they were expensive and/or difficult to access. In Bangladesh and in neighbouring India, the informal sector plugs the gap in state healthcare provision [19, 30]. Drug sellers and village doctors far outnumber the number of physicians, dentists and nurses in Bangladesh [19]. What Ahmed and colleagues call the "near total blindness to plurality at the policy level" ignores the reality that the informal sector is needed to meet the health needs of the populace. Restricting access to medicines through drug shops is both difficult (because they are numerous and operate outside of regulation) and dangerous because of the essential access they provide.

Further, several studies highlight that misuse of antibiotics is also widespread in formal provision [11, 31, 32] supported by unethical marketing practice [2, 15]. The households we visited with a current illness reported rapidly escalating provision of antibiotics from qualified doctors, including injected antibiotics, when symptoms did not quickly resolve. We cannot judge whether these uses were appropriate or not, but the descriptions given by families did not suggest careful diagnostic procedures before recommending further medication. It is therefore unclear the extent to which insisting on prescriptions would improve the appropriateness of antibiotics used. Finally, the common habit of discontinuing medicines when symptoms improve underlines the fact that antibiotics are not always consumed as prescribed.

The alternative to restricting access to antibiotics through drug shop, is to improve their practice. This has been tried in Bangladesh, although drug shop sellers' habits have proved difficult to shift [33]. Internationally, there is limited evidence so far about interventions to improve drug shop practice [15]. Miller and Goodman [15] suggest that increased knowledge alone is not the solution, because it will not overcome the need to maximise profit nor will availability decrease without regulation. In Bangladesh, regulation is being introduced through the model pharmacy and medicine shop initiative, which accredits only those which meet high standards of practice [34, 35]. In 2018 there were 193 such pharmacies and 154 shops [36], including one in the rural district where our research took place. These accredited providers provide a well regulated route to medicines, but their numbers are small. The Accredited Drug Dispensing Outlet provides a scalable model from Africa, which has allowed large numbers of drug shops to gain accreditation in Tanzania, Uganda and Liberia [37]. This may be a fruitful avenue for future intervention in other locations.

## Strengths and weaknesses

This study employed an international team with expertise across multiple fields (anthropology, psychology, microbiology, public health) to develop a rigorous method using a transparent approach to data interpretation. The sample included variation in location, income and household composition and included minority groups. A potential limitation of the study findings is in relation to healthcare seeking for animals, where the local government livestock services were highly effective and the provider of choice in the rural area which may have been location-specific. We believe the influence of this provision was large, and the good service these households received may not be typical of all areas. In the urban sample, we found few households with animals although previous research suggests a quarter of households in other high density urban areas had some animals in the compound [38]. In common with all qualitative studies, our study is diverse but small and is not designed to ensure generalizability. We provide thick, explanatory data for included households but accept they may not represent the experience of others.

## Conclusions

Drug shops provide an essential route to human medicines including antibiotics. Even where people also visit qualified private and public doctors, most often a drug shop will still provide the medicines. Drug shops in bazaars were operating as the first, intermediate and final points of access to medicines for most households. Great care should be taken before restricting access to antibiotics through this route. Qualified healthcare professionals provide only a small fraction of the antibiotics consumed and often rely on drug shops to supply prescribed medicines. While most people we spoke to could access qualified physicians, waiting times, cost and/or travel distance made these inaccessible for acute illness. Working to improve drug shop practice is likely to be more fruitful.

## Supporting information

**S1 File. Standards for Reporting Qualitative Research (SRQR) checklist.**
(DOCX)

**S1 Table. Characteristics of all participating households (HH).**
(DOCX)

## Acknowledgments

The authors would like to thank the participants in this study.

## Author Contributions

**Conceptualization:** Patricia Jane Lucas, Leanne Unicomb, Papreen Nahar, Mohammad Aminul Islam, Emily K. Rousham.

**Data curation:** Fosiul Alam Nizame.

**Formal analysis:** Patricia Jane Lucas, Mohammad Rofi Uddin, Nirnita Khisa, S. M. Salim Akter, Fosiul Alam Nizame.

**Funding acquisition:** Patricia Jane Lucas, Leanne Unicomb, Papreen Nahar, Mohammad Aminul Islam, Emily K. Rousham.

**Investigation:** Emily K. Rousham.

**Methodology:** Patricia Jane Lucas, Leanne Unicomb, Papreen Nahar, Fosiul Alam Nizame.

**Project administration:** Fosiul Alam Nizame, Emily K. Rousham.

**Writing – original draft:** Patricia Jane Lucas, Mohammad Rofi Uddin.

**Writing – review & editing:** Patricia Jane Lucas, Nirnita Khisa, Leanne Unicomb, Papreen Nahar, Mohammad Aminul Islam, Fosiul Alam Nizame, Emily K. Rousham.

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
