## [Decision Letter · Decision Letter 0]

31 Jul 2019

PONE-D-19-14635

Pathways to antibiotics in Bangladesh: A qualitative study investigating how and when low income households access antibiotics for humans or animals when they are ill

PLOS ONE

Dear Dr Lucas,

Thank you for submitting your manuscript to PLOS ONE. After careful consideration, we feel that it has merit but does not fully meet PLOS ONE’s publication criteria as it currently stands. Therefore, we invite you to submit a revised version of the manuscript that addresses the points raised during the review process.

We would appreciate receiving your revised manuscript by Sep 14 2019 11:59PM. To enhance the reproducibility of your results, we recommend that if applicable you deposit your laboratory protocols in protocols.io, where a protocol can be assigned its own identifier (DOI) such that it can be cited independently in the future. For instructions see: http://journals.plos.org/plosone/s/submission-guidelines#loc-laboratory-protocols

We look forward to receiving your revised manuscript.

Kind regards,

George Liu, PhD

Academic Editor

PLOS ONE

2. Please include copies of the interview guide(s) used in the study, in both the original language and English, as Supporting Information, or include a citation if they have been published previously.

Reviewers' comments:

Reviewer's Responses to Questions

**Comments to the Author**

1. Is the manuscript technically sound, and do the data support the conclusions?

Reviewer #1: Yes

Reviewer #2: Yes

Reviewer #3: Yes

2. Has the statistical analysis been performed appropriately and rigorously? 

Reviewer #1: N/A

Reviewer #2: N/A

Reviewer #3: I Don't Know

3. Have the authors made all data underlying the findings in their manuscript fully available?

Reviewer #1: Yes

Reviewer #2: Yes

Reviewer #3: Yes

4. Is the manuscript presented in an intelligible fashion and written in standard English?

Reviewer #1: Yes

Reviewer #2: Yes

Reviewer #3: Yes

5. Review Comments to the Author

Reviewer #1: The article is interesting as it was conducted qualitatively so that it can explore the depth of problems related to acquisition of medicines and antibiotics among low income households. However, there are some notes to improve the clarity of the article.

First, the title does not really reflect the contents. The pathways to antibiotics are not clear, as stated in line 463-6. It is suggested to include the term "medicines". So that, the title becomes something like "Pathways to medicines including antibiotics in Bangladesh: ...". Short title should also be changed with "... households in Bangladesh" or "...Bangladesh household"

Second, some grammatical errors occur.

a. Inconsistencies in writing quotes

Some quotes end with fullstop, but some do not and even have no unquote, such as shown in line 216, 232, 280, 334.

Some quotes put fullstop after unquote such as line 325, 328.

Some quotes have their respondent ID in brackets, while most do not, such as shown in line 391, 398, 401.

b. It seems that the article was prepared in hurry because some references are still blank or not formatted in Plos ONE Vancouver style. The cases are shown in line 58,521, 522, 544, 568.

c. Table 1 misses some lines.

d. References should be written in a correct, consistent manner. The journal names vary in style. Some are written in Title Case, such as reference 25; most are in Sentence case; and some others are unclear such as referencce 4, 17, 19. It's not necessarily to write the place of Lancet publisher (reference 11). Reference 1, 5, 6, and 7 are not written consistently. In addition, reference number 6 should be deleted or revised as the link is 404 (Page not found).

e. It is commonly preferable to write numbers <=10 as a word. For example, 7 is written as seven. The cases are in line 37, 157, 182, 253, 265.

f. It is also favourable to avoid the use of first-person pronouns, i.e. "we".

Third, in the beginning of the article the authors deligently write the full form of abbreviations in their first mention, but when in the middle part the authors forget to mention the full form of MBBS (line 193); TB (line 286); and icddr,b (line 287). Could MBBS be explained in more details?

Fourth, what is the justification to use only poor households. Do middle-income households may have different pathways? Both groups may have similar

preference if offered with low prices and easy access of drugs.

Fifth, in line 109 the authors mention the intention to include minority indigenous population. What criteria were used to ensure that the respondent is an indigenous person? Should they have indigenous maternal and paternal grandparents? What if a potential respondent has inter-racial or interethnic parents? WHat kind of ethnicities is dwelling in the research area?

Sixth, in line 119 are all members of the team able to communicate in English? Please provide more details.

Seventh, was the interview guide developed bilingual in English and Bangla? Please provide more details.

Eighth, in line 130 how did the researchers explain the concepts of antibiotics, side effects, and antimicrobial resistance during the interview? How could the

respondents differ antibiotics from other medicines or even vitamin and supplements? The information is better being written in methods section rather than in results.

Ninth, in line 136 why were only a third of recorded interviews translated into English? What criteria were used to select the records to be translated?

Tenth, what is the differences between drug shops and community pharmacies in the study? The reference 12 mentions "private pharmacies" instead of "drug shops". Are they similar? (line 70-8)

Eleventh, in line 525-6 the authors wrote "Eliminating direct access through 525 drug shops would not impact on this, and regulating to allow antibiotic access only via prescriptions would not necessarily improve practice." How could the authors create these claims? What evidence can be given? Some published articles about self-medication with antibiotics in developing countries can be incorporated in the discussion section.

Lastly, in line 535 the authors wrote "...lack of access to qualified doctors will push people into the black market." What does the term "black market" here mean? What evidence can be given?

Reviewer #2: Thank for this informative research that is highly needed.

Abstract:

I am not sure 'loose economic and regulatory system' is the correct term? There is regulation in Bangladesh, including the newly implemented 'model pharmacies' which promote prudent use of antibiotics. The issue is that the regulation is not enforced correctly.

In the methods of the abstract you should state the sampling method used to identify participants and why how saturation of data was achieved.

In the findings, rather than reporting 8 out of 13, perhaps it would be more appropriate as per chosen methods to discuss that antibiotics were reported to be one of the common therapies chosen for illness or words to that effect. It would be valuable to know if the households who opted for antibiotics were the more wealthy or this behaviour was reported across the different income categories.

The last sentence of the conclusion is important - as often access is so precarious and should not be restricted easily in order to tackle AMR. Would the authors please write this sentence more clearly, highlighting this issue in efforts to tackle AMR in LMICs.

Main manuscript:

Line 57 - sentence can be reworded, not sure impacts used in plural sense reads well

Line 58 - there is a plethora a papers which can be used for reference including the Lancet series on AMR and papers my Laxminarayan et al

Line 62 - and duration ...

line 73 Bangladeshis not Bangladeshi's

Line 80 Furthermore,....

Line 102 - Mirzapur district has an extensive surveillance programme for childhood infectious diseases that reaches thousands of households via the Child Health Research Foundation - See ANISA study in the Lancet

methods

So if only 1/3 of transcripts were translated is the data presented in this manuscript only from that batch or a wider selection was presented and translated for the purpose of the paper?

Did you do analysis of data as you gathered it? Did that also inform the saturation?

The first two sentences of results belong in the methods section

The drop out section is a bit confusing - can you break it down in urban vs rural and define the dropout for each?

It would help to provide the average income in professional urban and rural populations for Bangladesh, the difference between these two categories doesn't appear to be great

Line 183 - did you explore if there were specific conditions that they would seek help from 'fakirs' e.g. did infections feature in this? Or were there specific conditions they treat? It would be interesting in terms of delayed therapies for infections etc.

Line 265 - again i would refrain from giving numbers as in the context of the methods selected in this study they don't add anything. Better to focus on the findings and what they mean.

Did you explore whether the medicines for human use were ever given to the animals? Antibiotics specifically?

It would be useful and important to share the translated interview guide that was used to conduct the interviews - particularly when reading the section about knowledge and understanding under the pathways to antibiotics section of the results.

It would be good if in the discussion you can refer to the issue of access to antibiotics via informal routes, as described by some of your participants and the consequences of further restrictive regulation of this for poor households using some of the cases that you have followed up and provide details of in table 2. This is a critical point in the global efforts to tackle AMR, specifically in LMICs.

The sentence beginning with 'Qualified doctors may be...' in your conclusions needs refining - it is not clear who relies on drug sellers. The conclusion could be written more tightly. The last two sentences particularly. The last sentence - there are examples of this in Africa - the ADDOs (accredited drug dispensing outlets) - it would be good if this is referred to in the discussion. Also worth mentioning the model pharmacies in Bangladesh and the role these may have going forward - it is briefly alluded to but not explained in detail. The pharmacy of Kumudini hospital in Mirzapur for example is a model pharmacy.

Reviewer #3: Lucas et al present a qualitative assessment of access to antibiotics for human and animal use in Bangladesh. This is an interesting and thorough assessment of an important problem. It focuses on the two major problems relating to antibiotics--overuse and lack of access with a one health type approach.

I had a few minor comments:

introduction line 58 (ref) mentioned--add a reference

Line 71 replace dash with "to" in "eighty-ninety"

Line 80 what does "highly plural health provision" mean?

Table 1 has funny formatting of lines--carefully edit Left upper quadrant

Discussion line 568--again "(ref)" needs to be corrected

Line 574 "For humans" can be removed

6. PLOS authors have the option to publish the peer review history of their article (what does this mean?). If published, this will include your full peer review and any attached files.

Reviewer #1: Yes: Antonius Nugraha Widhi Pratama

Reviewer #2: Yes: ESMITA CHARANI

Reviewer #3: No

---

## [Author Response · Author response to Decision Letter 0]

16 Sep 2019

We have included in our submission a detailed rebuttal letter, addressing each of the reviewers comments. We thank the reviewers for their time and thoughtful comments.

---

## [Decision Letter · Decision Letter 1]

1 Nov 2019

Pathways to antibiotics in Bangladesh: A qualitative study investigating how and when households access medicines including antibiotics for humans or animals when they are ill

PONE-D-19-14635R1

Dear Dr. Lucas,

We are pleased to inform you that your manuscript has been judged scientifically suitable for publication and will be formally accepted for publication once it complies with all outstanding technical requirements.

With kind regards,

George Liu, PhD

Academic Editor

PLOS ONE

Additional Editor Comments (optional):

Reviewers' comments:

Reviewer's Responses to Questions

**Comments to the Author**

1. If the authors have adequately addressed your comments raised in a previous round of review and you feel that this manuscript is now acceptable for publication, you may indicate that here to bypass the “Comments to the Author” section, enter your conflict of interest statement in the “Confidential to Editor” section, and submit your "Accept" recommendation.

Reviewer #1: All comments have been addressed

2. Is the manuscript technically sound, and do the data support the conclusions?

Reviewer #1: Yes

3. Has the statistical analysis been performed appropriately and rigorously? 

Reviewer #1: N/A

4. Have the authors made all data underlying the findings in their manuscript fully available?

Reviewer #1: Yes

5. Is the manuscript presented in an intelligible fashion and written in standard English?

Reviewer #1: Yes

6. Review Comments to the Author

Reviewer #1: The authors have addressed all comments I gave. This research may contribute to the understanding on the access to antibiotics in economically deprived settings.

7. PLOS authors have the option to publish the peer review history of their article (what does this mean?). If published, this will include your full peer review and any attached files.

Reviewer #1: Yes: Antonius Nugraha Widhi Pratama

---

## [Editor Report · Acceptance letter]

14 Nov 2019

PONE-D-19-14635R1 

Pathways to antibiotics in Bangladesh: A qualitative study investigating how and when households access medicine including antibiotics for humans or animals when they are ill 

Dear Dr. Lucas:

I am pleased to inform you that your manuscript has been deemed suitable for publication in PLOS ONE. Congratulations! Your manuscript is now with our production department. 

With kind regards,

on behalf of

Dr. George Liu 

Academic Editor

PLOS ONE